# Use of Grape By-Products to Enhance Meat Quality and Nutritional Value in Monogastrics

**DOI:** 10.3390/foods11182754

**Published:** 2022-09-07

**Authors:** Cristina M. Alfaia, Mónica M. Costa, Paula A. Lopes, José M. Pestana, José A. M. Prates

**Affiliations:** 1CIISA—Centro de Investigação Interdisciplinar em Sanidade Animal, Faculdade de Medicina Veterinária, Universidade de Lisboa, 1300-477 Lisboa, Portugal; 2Laboratório Associado para Ciência Animal e Veterinária (AL4AnimalS), Faculdade de Medicina Veterinária, Universidade de Lisboa, 1300-477 Lisboa, Portugal

**Keywords:** grape by-products, feed ingredient, pig, poultry, meat parameters

## Abstract

Grape by-products could be used in monogastric animals′ nutrition to reduce feeding costs with conventional crops (e.g., maize and soybean meal) and to improve meat quality. The main grape by-products with the largest expression worldwide, particularly in the Mediterranean region, are grape pomace, grape seed, grape seed oil and grape skins. These by-products are rich sources of bioactive polyphenols, dietary fiber and polyunsaturated fatty acids (PUFA), more specifically, the beneficial *n*-3 PUFA, that could be transferred to pork and poultry meat. The potential biological activities, mainly associated with antimicrobial and antioxidant properties, make them putative candidates as feed supplements and/or ingredients capable of enhancing meat quality traits, such as color, lipid oxidation and shelf life. However, grape by-products face several limitations, namely, the high level of lignified cell wall and tannin content, both antinutritional compounds that limit nutrients absorption. Therefore, it is imperative to improve grape by-products’ bioavailability, taking advantage of enzyme supplementation or pretreatment processes, to use them as feed alternatives contributing to boost a circular agricultural economy. The present review summarizes the current applications and challenges of using grape by-products from the agro-industrial sector in pig and poultry diets aiming at improving meat quality and nutritional value.

## 1. Introduction

Current projections indicate an increase in demand for animal products, such as meat, until 2050, clearly driven by higher income and population growth [1]. Meat production is also estimated to double at an amount of 455 million tons (Mt) by 2050 [2]. Pork and poultry meat are the most consumed meats worldwide, and their increasing demand is due to low price, higher protein/lower fat content and consumer preferences shifting toward healthier meats [3]. Global consumption of pork and poultry meat is projected to increase to 127 Mt and 145 Mt, respectively, over the next seven years [4]. However, meat production requires feeding cereals and soybean meal to animals as main sources of energy and crude protein, respectively. The dependence of monogastric animals on these feed crops has becoming problematic due to food competition. In addition, the agro-industrial sectors produce about 1.3 billion tons of postharvest losses and processing by-products, which represents enormous environmental, social and economic costs every year [5]. However, most of these agro-industrial by-products are often underexploited and could be a promising alternative feed ingredient for the partial replacement of maize–soybean in monogastrics nutrition.

Grape (*Vitis vinifera)* is one of the most produced fruit crops, with an annual production of approximately 75 million tons, 41% of which are produced in Europe [6]. Almost 50% of the global grape production is addressed to vinification, a recognizable economic activity in Mediterranean countries [7,8]. The production of grapes in the Mediterranean region was 23 million tons in 2018 [9], with Italy, Spain and France standing out as the main producers. In those countries, the vineyards occupy a large area, for example, 13% in Spain, 10% in France, 9.2% in Italy and 2.5% in Portugal [10]. Grapes, together with olives, citrus and nuts, are the main crops with the largest expression in the Mediterranean area. As consequence, substantial quantities of grape by-products are generated on an annual basis, which represents a big challenge for waste management [8].

Grape pomace, the major by-product of the grape industry, is a solid organic waste residue obtained after the juice extraction from grapes, consisting of pressed grape pulp and skins, as well as grape seeds and stems. It is estimated that around 20–25% of the total weight of grapes crushed for wine production results in grape pomace [11,12]. Grape stems, although least applied, are removed before winemaking and represent around 5% of wine by-products [13]. Occasionally, stem is used to extract grape seed oil [14,15]. Even though grape by-products represent environmental concerns, they are a rich source of bioactive substances, such as polyphenols, which are well-known for their antioxidant, anti-inflammatory, anticarcinogenic, cardioprotective and vasodilatory properties [7,16,17].

Therefore, the goal of this systematic review was to assess the available literature on the use of grape by-products (red and white grape pomace, grape seed, grape seed oil and grape skin), as feed supplements or ingredients, and their impact on pork and poultry meat quality and nutritional value. The methodology applied to the literature search was based on reference databases, such as PubMed (NCBI, Bethesda, MD, USA), Web of Science (Clarivate Analytics, Philadelphia, PA, USA) and ScienceDirect (Elsevier, Amsterdam, The Netherlands), using “grape by-products”, “meat quality”, “pigs” and “poultry” as keywords.

## 2. Nutritional Composition of Grape By-Products

Grape by-products are a relatively good source of valuable nutrients and biologically active substances with potential health-promoting effects, such as phenolic compounds. These compounds are a heterogeneous group of phytochemicals comprising flavonoids, phenolic acids, tannins (hydrolysable and the nonhydrolyzable or condensed tannins), stilbenes, anthocyanins, xanthines and lignans [18]. Table 1 shows the chemical and nutritional composition of grape by-products. The nutrient variability of grape by-products depends on the grape cultivar, maturity level, environmental factors and processing conditions [19].

Grape pomace is particularly rich in dietary fiber, ranging from 51.4 to 83.6% of dry matter, with significant differences between insoluble and soluble fractions from white or red grape pomace [43]. The insoluble fraction of grape pomace accounts for 61.3 and 73.5%, whereas the soluble fraction comprises 10.3 and 3.7% of the total fiber in white and red grape pomace, respectively. The insoluble fraction is mainly composed of cellulose, hemicellulose and lignin, and the soluble is mostly constituted by uronic acids [44]. Grape pomace also contains significant amounts of proteins, carbohydrates, lipids, vitamins, minerals and a wide diversity of polyphenols. The latter include phenolic acids and phenolic alcohols, flavonoids (catechin, epicatechin, quercetin-3-O-rhamnoside and luteolin), stilbenes (resveratrol) and proanthocyanidins [15,23]. Most of the available studies concerning phenolic profile are mainly dedicated to pomace derived from red grape varieties instead of white varieties [16]. Although grape phenolic compounds are responsible for the color, astringency, flavor and aroma of wine [45], they also display strong antioxidant and antimicrobial effects against various pathogenic microorganisms [46]. The antioxidant activity of these compounds allows the efficient removal of superoxide anions, hydroxyl and lipid peroxyl radicals, minimizing oxidative reactions [47,48,49]. The beneficial effects of polyphenols underlie the use of grape pomace in animal feed [50]. Regarding fatty acid profile, linoleic acid (18:2*n*-6) and oleic acid (18:1*c*9) are the major unsaturated fatty acids of grape pomace. As shown in Table 1, grape skin is particularly rich in polyphenols, such as anthocyanins, catechins and flavonols [33], especially from red grapes [51]. Grape seed is predominantly composed of 40% of fiber (60–70% of nondigestible and 29% of complex carbohydrates), around 11% of protein, 13–19% of fat (rich in essential fatty acids), as well as minerals and nonphenolic compounds (tocopherols and β-carotene) with high antioxidant activity, thus inhibiting lipid peroxidation in biological membranes [52]. Grape seed also contains extractable phenolic antioxidants, such as phenolic acid, flavonoids, procyanidins and resveratrol [11]. Grape seed extract and grape seed oil are by-products derived from the grape seeds used for grape juice and wine processing. The grape seed extract, aqueous or alcoholic, is extracted, dried and purified to yield a polyphenolic compound-enriched extract [53]. Grape seed oil contains high amounts of lipophilic (vitamin E, unsaturated fatty acids and phytosterols) and hydrophilic phenolic compounds (flavonoids, carotenoids, phenolic acids and tannins) [35,39]. The nonphenolic antioxidants occur in grape seeds but are more concentrated in grape seed oil [37,38]. The content of vitamin E homologues in grape seed oils ranges from 499–1575 mg/kg of γ-tocotrienol, 85.5–578 mg/kg of α-tocopherol and 69–319 mg/kg of α-tocotrienol [37,40]. Concerning fatty acid profile, the predominant fatty acids of grape seed oil are linoleic acid (18:2*n*-6, 60.1–73.1%) and oleic acid (18:1*n*-9, 13.7–26.5%), as unsaturated fatty acids, followed by the saturated fatty acids (SFA), palmitic acid (16:0, 6.5–9.7%) and stearic acid (18:0, 3.50–7.30%) [54,55]. For example, the Cabernet Sauvignon and Royal Rouge grapes have 60.9–64.4% of polyunsaturated fatty acids (PUFA) and a high ratio of PUFA/SFA [32]. These grape seed by-products exhibit beneficial bioactive properties, such as antioxidant, anti-inflammatory, antimicrobial and anticancer properties [56,57].

In addition to their interesting nutritional properties, grape by-products also contain minor proportions of antinutritional compounds, mainly fiber, procyanidins (condensed tannins) and phytic acid [23], which limit their use in poultry and pig diets. Grape skin is particularly rich in dietary fiber (74% of weight), mostly composed of hemicelluloses [58] and polyphenols [59], while grape stem is fully composed by tannins, up to 50% of total polysaccharides [60]. However, grape by-products can be converted to more effective products by exogenous enzymes supplementation or pretreatment methods, such as polyethylene glycol treatment and fermentation, before incorporation into monogastric diets [61]. Polyethylene glycol partially inactivates grape pomace-condensed tannins [62]. The fermentation process could improve the nutritional composition of these by-products, eliminating the antinutritional compounds and increasing the antioxidant and antimicrobial effects by raising the phenolic compounds amount [23]. Moreover, the phenolic compounds in grape by-products have free radical scavenging activities and, thus, terminate the radical chain reactions involving lipid and protein oxidative reactions [63], which modify nutritional and sensorial properties of meat and meat products. Lipid oxidation promotes meat off-flavors while myoglobin oxidation and metmyoglobin formation are responsible for color changes [64].

## 3. Effects of Dietary Grape By-Products on Pork Quality and Nutritional Value

Meat quality is a key factor in consumer preference. The major meat quality parameters include water holding capacity (WHC), color, oxidative stability, tenderness, flavor and shelf life. Furthermore, fatty acid composition of meat, due to its relation to human metabolic disorders, are among the factors usually used for evaluating the nutritional and healthy values of meat [65]. Table 2 shows the influence of dietary incorporation of grape by-products on pork-quality traits. Although different concentrations and experimental periods were reported, feeding pigs with grape pomace and grape seeds enriched in dietary fiber and phenolic compounds has a positive impact on fatty acid profile and reduces, in general, the susceptibility of pork to oxidation [66,67].

The supplementation of 3% of grape pomace fermented by *Saccharomyces boulardii* in finishing pigs for 56 days increased pork color parameters a* (redness) and b* (yellowness) and decreased thiobarbituric acid reactive substances (TBARS) [68], the usual method to quantify malondialdehyde (MDA), which is the major marker of lipid oxidation. The reduction of lipid oxidation, determined as TBARS, could be related to the availability of phenolic antioxidants that inhibit lipid oxidation [76]. In turn, Habeanu et al. [69] found that the supplementation of 5% of dried grape pomace for 28 d in pigs had no effect on SFA, MUFA and PUFA. Nevertheless, grape pomace at this level increased *n*-3 PUFA and decreased arachidonic acid (20:4*n*-6) with a contextual decrease in the *n*-6/*n*-3 ratio. Beyond polyphenols, grape pomace had the highest proportions of PUFA, particularly regarding alpha-linolenic acid (18:3*n*-3, ALA), the precursor of the health-beneficial eicosapentaenoic (20:5*n*-3, EPA) and docosahexaenoic (22:6*n*-3, DHA) acids. Similarly, Kafantaris et al. [71] reported that 9% of red grape pomace fed to piglets for 30 d had no effect on SFA and MUFA, but increased PUFA, mainly ALA, EPA and DHA. This increase in *n*-3 PUFA could be associated with the probiotic effect of grape pomace [77]. Trombetta et al. [72] also reported higher proportions of *n*-3 PUFA in pork, with a concomitant decrease in the *n*-6/*n*-3 ratio, when pigs were fed with 3.5 and 7% of ensiled grape pomace (as a source of natural antioxidants) with linseed oil (as a source of *n*-3 fatty acids) for 86 d. The inclusion of ensiled grape pomace with linseed oil in finishing pig diet generated meat with higher nutritional value (increased PUFA/SFA ratio and reduced *n*-6/*n*-3 ratio). However, the incorporation of grape pomace in pigs’ diets, especially at the highest inclusion level of 7%, increased lipid oxidation. The possible antioxidant effect of ensiled grape pomace could be limited to the initial storage period. In agreement, Bertol et al. [70], using grape pomace in swine finishing diet, did not observe a protective effect on muscle fatty acids but instead an improvement of meat color by increasing redness and color saturation, which suggests a potential antioxidant effect of grape pomace. 

O’ Grady et al. [73] observed that the oxidative stability (TBARS) in raw and cooked *longissimus dorsi* (LD) muscle of pigs was not enhanced by dietary grape seed extract, but increasing concentrations (700 mg/kg), on days 12 and 16 of storage at 4 °C, reduced lipid oxidation to a minor extent in raw LD muscle compared to the control. More recently, Xu et al. [74] investigated the supplementation effects of dietary grape seed proanthocyanidin extract on meat quality, muscle fiber characteristics and antioxidant capacity of finishing pigs. The former authors found that pH45min, L* (lightness) and b* values, drip loss at 24 h, cook loss and marbling score were unaffected; however, total PUFA and PUFA/SFA ratio increased by dietary grape seed proanthocyanidin extract supplementation. Although dietary supplementation did not change the *n*-6/*n*-3 ratio, the enrichment in *n*-3 PUFA (mainly ALA and EPA) may improve the nutritional and healthy values of pork. 

The supplementation of pig diets with red wine solids has no effect either on meat quality nor on its oxidative stability [75].

## 4. Effects of Dietary Grape By-Products on Poultry Meat Quality and Nutritional Value

The inclusion of grape by-products in poultry diet and their impact on meat quality traits is depicted in Table 3. The incorporation levels in poultry are up to 10% in feed, as for pigs.

Kasapidou et al. [29] described that supplementation of 0.25, 0.5 and 1% of grape pomace had no effect on breast muscle color (L* and b*), but grape pomace at 1% decreases a* values. Moreover, Aditya et al. [78] observed that 0.5, 0.75 and 1% of grape pomace had no effect on L* (at 5 and 10 d of storage) and b* (at 5 d of storage) but decreased a* (at 5 and 10 d of storage) and b* (at 10 days of storage). Similar results were obtained by Bennato et al. [79], who found that grape pomace at 2.5, 5 and 7% had no effect on L* but increased a* and b* values. The dietary grape pomace incorporation at the 5 and 10% level in male broiler chicks for 21 d reported an increase in total PUFA [80]. Moreover, the incorporation of 5 and 7% of grape pomace during 49 d in male chickens increased PUFA, decreased SFA and had no effect on MUFA [79]. Most of the studies performed, so far, reported that the inclusion of grape pomace (from 0.01% to 7% feed) in broiler chicks, chickens and ducklings have either no effect [29,83] or reduce TBARS values in breast and thigh [78,79,80,81,82,86]. Jurčaga et al. [83] also reported no protective effect of dietary red grape pomace against lipid oxidation, which agrees with those observed by Kasapidou et al. [29]. In turn, broiler chicks fed diets with red grape pomace up to 3% decreased lipid oxidation in breast and thigh stored for 1, 4 and 7 d [81]. Likewise, Brenes et al. [82] observed that dietary inclusion of red grape pomace up to 6% was able to delay lipid oxidation in the breast of broiler chicks stored for 1, 4 and 7 d. This positive impact on meat oxidative stability could be due to the presence of polyphenols in grape pomace with antioxidant potential, thus providing protection against lipid oxidation in tissues [82]. Some variations in antioxidant activity of grape pomace might be related to its own composition (stems, seeds, skin) and processing methods (e.g., drying and heating) [87]. The supplementation of red and white grape pomace in polyunsaturated fatty acids-enriched broiler diets (4% flaxseed meal) led to improved meat color and oxidative stability, especially in thigh [84].

The incorporation of grape seeds at 2% into broiler’s diet for 5 wk also increased PUFA (13.4%), mostly *n*-3 PUFA (10.0%), and reduced SFA and MUFA amounts [85], as happened with grape pomace [80]. Furthermore, Romero et al. [12] observed that the combination of grape seed and grape skins in broiler chickens’ diet resulted in lower meat lipid oxidation than the diets including grape seeds or grape skins separately. Thus, the antioxidant potential effectiveness of grape by-products depends on the nature and the total polyphenol content incorporated into the diet. 

Grape seed extract at 0.01 and 0.02% levels in feed of female Pekin ducklings also decreased TBARS [86].

Together, the main effects of the addition of grape by-products to pigs and poultry diets up to 10% are shown in Figure 1, indicating a higher nutritional value and oxidative stability of pork and poultry meat, mainly attributed to dietary fiber and polyphenols.

## 5. Conclusions and Future Directions

This review highlighted the use of dietary grape by-products to replace conventional feedstuffs in pig and poultry diets aiming at enhancing meat quality and nutritional value. Grape pomace, followed by grape seeds, have been the main by-products used as feedstuffs in monogastric animals. These by-products are good sources of valuable nutrients, mainly fibers, and bioactive substances, such as phenolic compounds, with potential health-promoting effects. 

The dietary inclusion level of grape by-products found in the reviewed literature is up to 10% in both pigs and poultry. In general, feeding pigs with grape pomace and grape seeds have a beneficial influence on pork fatty acid profile (increase in *n*-3 PUFA and decrease in *n*-6/*n*-3 ratio) and reduce its susceptibility to oxidation (increase in polyphenols). In addition, most studies reported here showed that the incorporation of grape pomace in poultry diets have either no effect or an antioxidant effect in breast and thigh meats. Summing up, there are clear benefits of using grape by-products as functional feed supplements or ingredients (<10%) by improving meat color, fatty acid composition and extending meat shelf life.

However, using these by-products as nutrient sources poses a major challenge due to the presence of small amounts of antinutritional factors and their consequently negative effects on the monogastric digestive systems. In view of this constraint, further studies are warranted to clarify the appropriate inclusion level for each grape by-product in monogastric diets and to develop effective pretreatments to solve this indigestibility limitation.

## Figures and Tables

**Figure 1 foods-11-02754-f001:**
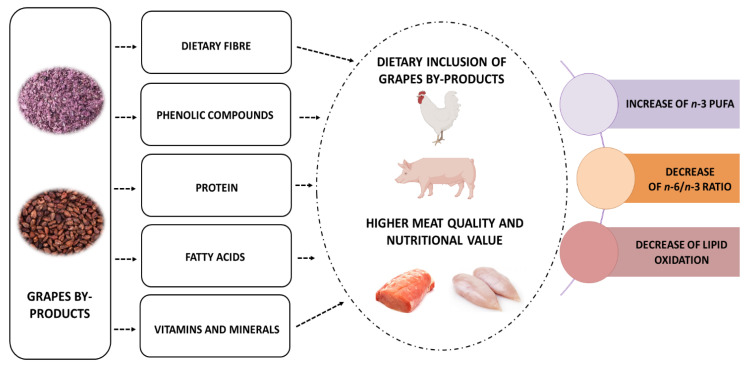
Major impacts of feeding poultry and swine with grape by-products on meat quality.

**Table 1 foods-11-02754-t001:** Chemical and nutritional composition of the main grape by-products (values are expressed on a dry weight basis, w/dw).

Item	Grape Pomace ^1^	Grape Skin ^2^	Grape Seed ^3^	Grape Seed Oil ^4^
Crude protein (%)	8.90–13.9 (12.3) ^†^	6.54–13.8 (10.2)	6.00–12.7 (9.55)	–
Fiber				
Crude fiber (%)	14.3–74.5 (38.9)		45.8–47.4 (46.6)	–
ADF (%)	32.3–48.4 (40.4)	19.3–49.0 (34.2)	–	–
TDF/NDF (%)	40.9–59.1 (48.8)	17.3–56.3 (36.8)	45.2	–
SDF (%)	2.35	0.72–1.72 (1.22)	79.9	–
ADL/Lignin (%)	18.2–42.5 (29.8)	28.3–43.7 (36)	ND	–
Crude fat (%)	2.12–13.5 (7.9)	1.14–6.33 (3.74)	4.82–20.7 (12.7)	–
Sugar (%)	2.10–14.2 (6.4)	4.90–14.6 (9.75)	ND	–
Ash (%)	2.40–23.7 (13.1)	2.53–7.59 (5.06)	2.60–3.88 (3.18)	–
Moisture (%)	3.39–10.2 (7.2)	7.00–19.9 (13.5)	6.53–7.60 (7.07)	–
Fatty acid profile (% total fatty acids)				
16:0	12.0–13.4 (12.7)	–	8.52	6.5–9.7 (8.1)
18:0	4.31–5.07 (4.69)	–	3.95	2.84–7.30 (4.49)
20:0	0.57–0.84 (0.71)	–	0.10	0.14–0.16 (0.15)
16:1*n*-7	0.05–0.10 (0.08)	–	0.32	0.08
18:1*n*-9	12.2–14.0 (13.1)	–	22.9	13.7–26.5 (20.1)
20:1*n*-9	0.03–0.04 (0.04)	–	0.09	0.00–0.97 (0.39)
18:2*n*-6	58.0–62.7 (60.4)	–	62.5	60.1–74.7 (66.0)
18:3*n*-3	1.7–2.8 (2.25)	–	0.35	0.00–0.87 (0.42)
SFA	20.6–21.8 (21.2)	–	13.7	10.4–11.7 (13.1)
*cis*-MUFA	14.3–15.4 (16.5)	–	23.3	14.8–18.7 (16.7)
PUFA	60.9–64.4 (62.7)	–	62.9	68.3–74.9 (71.6)
*n*-3 PUFA	1.7–2.8 (2.25)	–	0.35	0.20
*n*-6 PUFA	58.1–62.7 (60.4)	–	62.5	74.7
Mineral composition				
Macrominerals (g/kg)				
Ca	3.2–7.0 (5.1)	41–70 (76)	4.7–7.0 (5.85)	–
K	15.6–26.5 (21.1)	17.9–24.7 (21.3)	8.32–33.1 (20.7)	–
Mg	0.80–0.90 (0.85)	0.40–0.83 (0.62)	1.30–1.79 (1.55)	–
P	2.0–3.6 (2.8)	23–29 (26)	0.83–23.7 (12.3)	–
Microminerals (mg/kg)				
Cu	12.4–387 (199)	23–124 (73.5)	<10.0–73.3	–
Fe	64–185 (124.5)	117–398 (257.5)	45–120 (82.5)	–
Mn	13–17 (15)	13–17 (15)	16.5–27.5 (22)	–
Zn	12 –18 (13)	12–18 (13)	18.2–26.9 (22.6)	–
Vitamin E homologues (mg/kg)				
α-Tocopherol	–	–	–	85.5–578 (331.8)
γ-Tocopherol	–	–	–	–
δ -Tocopherol	–	–	–	–
α-Tocotrienol	–	–	–	69–319 (194)
γ-Tocotrienol	–	–	–	479–1575 (1027)
Phenolic compounds				
Total phenols (mg GAE/g)	12.3–58.9 (27.9)	9.70–52.3 (31)	261.3 *	–
Total anthocyanin (mg Mvd-3-glu/g)	1.3–3.4 (2.0)	0.29–1.42 (0.86) **	ND	–
Total tannins (mg TAE/g)	107.2	44.9–73.0 (59)	33.9	–
Total flavonoids (mg CE/g)	26.9	31.0–61.2 (46.1) ***	ND	–

^†^ Hyphenated results are ranges followed by average values. ADF, acid detergent fiber; TDF, total dietary fiber; NDF, neutral detergent fiber; SDF, soluble dietary fiber; ADL, acid detergent lignin; SFA, saturated fatty acids; MUFA, monounsaturated fatty acids; PUFA, polyunsaturated fatty acids; GAE, gallic acid equivalents; Mvd-3-glu, malvidin-3-glucoside; TAE, tannic acid equivalents; CE, catechin equivalents; ND, not detected. * Value is expressed as epicatechin equivalents; ** Only for skins of red grape pomace; *** Total flavonols. Supporting literature: ^1^ [15,20,21,22,23,24,25,26,27,28,29,30,31,32], ^2^ [15,33], ^3^ [15,19,34,35,36], ^4^ [35,37,38,39,40,41,42].

**Table 2 foods-11-02754-t002:** Main effects of dietary inclusion of grape by-products on pork quality traits.

Grape By-Product	Incorporation Level in Feed (% Dry Matter)/Experiment Duration	Initial Weight/Age of Animals	Main Findings	References
Grape pomace	3% for 105 d	Pigs at 19.3 kg and 21-d-old	- No effect on pH, marbling, firmness and L* - Increase in a* and b* (20% and 31%, respectively) - Decrease in TBARS (47%)	[68]
	5% for 28 d	Hybrid pigs at 75.5 kg	- No effect on SFA, MUFA, PUFA and *n*-6 PUFA - Increase in 18:3*n*-3 and *n*-3 PUFA (26% and 22%, respectively) - Decrease in 20:4*n*-6 and *n*-6/*n*-3 ratio (29% and 11%, respectively)	[69]
	3–5% for 21 d and 6–10% for 17 d	Barrows at 80.0 kg	- No effect on pH45min, pH24h, L* and b*, cooking loss and TBARS - Decrease in a* (18% with 3–5% dosage)	[70]
Red grape pomace	9% for 30 d	Piglets at 4.8 kg and 20 d old	- No effect on SFA, MUFA and *n*-6 PUFA - Increase in ALA (19%), EPA (63%), DHA (56%), PUFA (13%) and *n*-3 PUFA (50%) - Decrease in *n*-6/*n*-3 ratio (46%)	[71]
	3.5 and 7% for 86 d	Castrated males and female pigs at 48.6 kg and 180 d old	- No effect on pH, drip loss, color marbling, WHC, cooking loss, lipid, moisture, protein, ash, color, shear force, cholesterol, MUFA, n-6 PUFA and sensory parameters (tenderness, juiciness and off-flavor) - Increase in TBARS (up to 85%), PUFA (up to 31%), *n*-3 PUFA (up to 88%) and PUFA/SFA ratio (38%) - Decrease in SFA and *n*-6/*n*-3 ratio (8% and 13%, respectively)	[72]
Grape seeds	1% of grape seeds and 5% of flax meal for 42 d	Hybrid pigs with an average weight of 60.2 kg	- No effect on crude protein, dry matter, ash, SFA, MUFA, PUFA and PUFA/SFA ratio - Increase in *n*-3 PUFA (38%)	[31]
Grape seed extract	0.01, 0.03 and 0.07% for 56 d	Male and female pigs with an average body weight of 46.0 kg	- No effect on pH, TBARS and color parameters	[73]
	0.005, 0.01 and 0.02% for 49 d	Pigs with an average body weight of 67.5 kg	- Increase in pH24h and redness (3% and 15%, respectively) - Increase in crude protein (7%), PUFA (20%), *n*-3 PUFA (13%) contents and PUFA/SFA ratio (26%) - Decrease shear force and drip loss at 48 h (4% and 39%, respectively)	[74]
Red wine solids	0.36% for 56 d	Male and female pigs at 74.0 kg and 3 months	- No effect on fatty acid composition, pH, color, drip loss, cooking loss, shear force, ash, crude protein and TBARS	[75]

**Table 3 foods-11-02754-t003:** Main effects of dietary inclusion of grape by-products on poultry meat quality traits.

Grape By-Product	Level in the Diet (% dry matter) and Experiment Duration	Initial Weight and Age of Animals	Main Results	References
Grape pomace	0.25, 0.5 and 1% for 42 d	Broiler chicks at 1 d old	- No effect on L*, b* and TBARS - Decrease in a* (around 15% at 1% dosage)	[29]
	0.5, 0.75 and 1% for 28 d	Broiler chicks at 3 d old	- No effect on L* (at 5 and 10 d of storage) and b* (at 5 d) - Decrease in TBARS at 0 d (up to 23%), 5 d (up to 34%) and 10 d (up to 45%) of storage - Decrease in a* at 5 d (up to 61%) and 10 d (up to 56%) and b* at 10 d (up to 7%) of storage	[78]
	2.5, 5 and 7% for 49 d	Male chickens at 1 d old	- No effect on pH48h, cooking loss, L*, moisture, dry matter, total lipids, proteins, ash and MUFA - Increase in drip loss (20% with 7% dosage); a* and b* (up to 69% and 34%, respectively, in all dosages) - Increase in 18:2*n*-6 and PUFA (up to 12% and 11% with 5% and 7% dosages) and PUFA/SFA ratio (21% with 5% dosage) - Decrease in SFA (up to 11% with 5% and 7%) - Decrease in TBARS (up to around 44% and 51% with 5% and 7%, respectively, after 3 and 7 d storage at 4 °C)	[79]
Red grape pomace	5 and 10% for 21 d	Male broiler chicks with an average weight of 591 g at 21 d old	- Increase in PUFA (up to 32% with 5 and 10% dosages) and α-tocopherol (47% with 10% dosage) - MUFA (up to 31%) (5% and 10% dosages) and SFA (15% with 10% dosage) - Decrease in TBARS (up to 33% and 48% with 5 and 10% dosages after 1 and 4 d, respectively)	[80]
	0.5, 1.5 and 3% for 21 d	Male broiler chicks at 1 d old	- Decrease in TBARS in breast after 4 d (up to 33%) and 7 d (up to 47%) and in thigh after 7 d (up to 30%) of refrigerated storage	[81]
	1.5, 3 and 6% for 21 d	Male broiler chicks at 21 d old	- Decrease in TBARS in breast after 1 (up to 41%), 4 (up to 28%) and 7 d (up to 36%) of refrigerated storage	[82]
	1, 2 and 3% for 42 d	Hybrid broiler chickens at 1 d old	- No effect on TBARS	[83]
Red and white grape pomace	3% or 6% for 28 d(14–42 d)	Broiler chickens with an average weight of 506 g at 14 d old	- Positive effect on meat color and texture - Decrease in TBARS in breast up to 18% (3% white and 6% red) and in thigh up to 35% (white) and to 26% (red) (3% and 6%)	[84]
Grape seeds	2% for 35 d(14–49 d)	Broilers with average weight of 312 g at 14 d old	- No effect on dry matter, crude protein and *n*-6/*n*-3 ratio - Increase in UFA (2% in breast and thigh), PUFA (13% in breast and 16% in thigh), *n*-3 PUFA (10% in breast and 19% in thigh) and *n*-6 PUFA (14% in breast and 15% in thigh) - Decrease in SFA (5% in breast and thigh) and MUFA (8% in breast and 7% in thigh)	[85]
Grape seed extract	0.01 and 0.02% for 42 d	Female Pekin ducklings at 52.0 g and 1 d old	- No effect on pH45min and color - Increase on pH24h WHC (up to 4%) and to 9%, respectively) - Decrease in cooking loss (up to 13%), TBARS (up to 25%) and drip loss (up to 9% and to 4% after 3 and 5 d, respectively)	[86]

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
