# Peer review of "Use of Grape By-Products to Enhance Meat Quality and Nutritional Value in Monogastrics"

_foods, 2022, doi:10.3390/foods11182754_

Round 1

Reviewer 1 Report

The main question addressd by the research is to asses the available literature on used of grape by-products as ingredient in pig and poultry feed and reported their influence on the meat quality. The article is interesting because focus on a great waste problem of wine food chain and explain how the by-products could enter in circulating economy concept if they are used to animal feed. The topic is original because reviewer the literature about this topic and highlighted how is need to conduct further studies to improving the right level of by-product inclusion in  feed, for monogastric, to avoid the negative aspects of these grape by-products. For me represent an overview about this argument, that highlight the positive and negative aspects of the subject.

The paper is well written. The text is clear and easy to read. Conclusions are consistent with the evidence and arguments presented, authors have been summarize how the grape by-product have been influence the meat quality, by the literature used, and reported the future challenges to improve the way to use these by-products. They address the main question posed.

Dear Authors I found your work is very interesting and actuality. I suggest only to conform the header of the tables 2 and 3; and put in subscript format  the numeber "48" of the pH48 in the first row of table 3.

In biblography citation n.20  please insert all authors and delete et al.,.

Author Response

Comments and Suggestions for Authors

The main question addressed by the research is to assess the available literature on used of grape by-products as ingredient in pig and poultry feed and reported their influence on the meat quality. The article is interesting because focus on a great waste problem of wine food chain and explain how the by-products could enter in circulating economy concept if they are used to animal feed. The topic is original because reviewer the literature about this topic and highlighted how is need to conduct further studies to improving the right level of by-product inclusion in feed, for monogastric, to avoid the negative aspects of these grape by-products. For me represent an overview about this argument, that highlight the positive and negative aspects of the subject.

The paper is well written. The text is clear and easy to read. Conclusions are consistent with the evidence and arguments presented, authors have been summarize how the grape by-product have been influence the meat quality, by the literature used, and reported the future challenges to improve the way to use these by-products. They address the main question posed.

Dear Authors I found your work is very interesting and actuality. I suggest only to conform the header of the tables 2 and 3; and put in subscript format the number "48" of the pH48 in the first row of table 3. In bibliography citation n.20, please insert all authors and delete et al.,.

Reply: Thanks for your comments. We acknowledge the reviewer´s suggestion. The authors reformulated the header of the table 3 to match with that of table 2. In addition, the authors also reviewed the reference 20 that is now reference 44. We decided to indicate quality traits, such as pH48h, in normal text format in the revised document.

Reviewer 2 Report

The manuscript is interesting, it is necessary to attend to the following observations:

Line 18: keywords are additional words that help increase the search for the manuscript in the database, so it is recommended to use words that are not included in the title of the work. It is recommended to modify some repeated words.

Line 33: could indicate the average per capita consumption of both types of meat worldwide

Line 42: modify…maize-soybean

Line 48: delete parenthesis

Line 120: the table must appear immediately after the first paragraph in which it is mentioned

Line 96: modify…[24–26]

Line 99: 13–19%

Line 116: 60.9–64.4%

Line 121: missing information related to skin

Line 121: modify…8.9–13.9; homogenize format in the value ranges

Line 121: Total anthocyanin (mg ?????/g)

Line 121: Total flavonoids (mg ????/g)

Line 127-132: delete authors names

Line 145-143: It could be interesting to include the effect of these by-products on protein oxidation and metmyoglobin formation.

Line 145: It could be interesting to include a brief section where the parameters to evaluate meat quality are included or what they consist of, so that the reader who is not an expert in the subject, can follow up on the results mentioned later in the manuscript

Line 193: the table must appear immediately after the first paragraph in which it is mentioned

Line 152: indicate the meaning of a* and b* parameters

Line 156,163,166: delete authors names

Line 157: modify…on SFA, MUFA and PUFA.

Line 157: modify…28 days by 28-d or 28 d; abbreviate from this section throughout the document

Line 164:…but increased PUFA

Line 174,175,178,182: delete authors names

Line 176: indicate the meaning of MDA

Line 184: indicate the meaning of L* parameter

Line 193: it is necessary delete the authors names of references included in the table

Line 193: in the reference Yan and Kim [62],…in which percentage the color values were increased, as well as the reduction in the oxidation of lipids respect to the control

Line 193: in the references Habeanu et al. [64] and Bertol et al. [68], the animal age was not indicated

Line 193: in the reference Bertol et al. [68], modify…..(3–5%)..(6–10%)

Line 193: in the reference Kafantaris et al. [65], the animal weight was not indicated

Line 193: in the references [64,65,67], by what percentage did the values increase or decrease?

Line 193: in the references [67], indicate the meaning of WHC, and indicate the sensory traits

Line 193: in the reference [50], the animal age was missed and indicate the percent increase respect to the control

Line 193: in the reference [70], the animal age was missed

Line 193: in the reference [71], the animal age was missed and indicate the percent increase or decrease respect to the control

Line 199: delete authors names; it is necessary to make changes through the manuscript

Line 200:… muscle colour (L* and b*), but….

Line 201: …decrease a* values.

Line 212: [73–75,77–79].

Line 225: …5 wk also…

Line 234: indicate the percentage of increase or reduction in the values that presented a significant effect with respect to the control

Line 234: in some references the weight of the animals is omitted, could you specify if the value was not reported in the cited reference, if it is not available

Line 234: in the reference [74], pH48h

Line 248: 1563815678.

Line 287: 217260.

Line 293: 221237.

Line 297: 152157.

Line 302: 117.

Line 304: 1157911587.

Line 306: 16071615.

Line 308: 267337.

Line 310: 24732505. ….review the page range format in the following references

Author Response

Comments and Suggestions for Authors

The manuscript is interesting, it is necessary to attend to the following observations:

Reply: Thanks for your comments. We acknowledged the reviewer´s suggestions.

Line 18: keywords are additional words that help increase the search for the manuscript in the database, so it is recommended to use words that are not included in the title of the work. It is recommended to modify some repeated words.

Reply: We thank the reviewer for this suggestion. Some keywords have now been included (please see page 1, line 27).

Line 33: could indicate the average per capita consumption of both types of meat worldwide

Reply: Thank you for this suggestion. We have provided this information in the revised form of the manuscript (please see page 1, lines 36-37).

Line 42: modify…maize-soybean

Reply: Corrected, as suggested by the reviewer.

Line 48: delete parenthesis

Reply: Corrected, as suggested by the reviewer.

Line 120: the table must appear immediately after the first paragraph in which it is mentioned

Reply: The reviewer is absolutely right. The authors re-introduced the Table 1 after the first paragraph in which it is mentioned (please see page 2, lines 88-89).

Line 96: modify…[24–26]

Reply: Corrected, as suggested by the reviewer. The number references are now [47-49] in the revised manuscript.

Line 99: 13–19%

Reply: Corrected, as suggested by the reviewer.

Line 116: 60.9–64.4%

Reply: Corrected, as suggested by the reviewer. The values little changed because we added some references in Table 1.

Line 121: missing information related to skin

Reply: The information related to skin is now added in Table 1.

Line 121: modify…8.9–13.9; homogenize format in the value ranges

Reply: Thank you for this. The authors carefully reviewed the value ranges throughout the manuscript.

Line 121: Total anthocyanin (mg ?????/g)

Reply: Total anthocyanins are expressed as mg of malvidin-3-glucoside/g of by-product. This information is now added in Table 1 of the revised manuscript

Line 121: Total flavonoids (mg ????/g)

Reply: Total flavonoids are expressed as mg of catechin equivalents This information is now added in Table 1 of the revised manuscript.

Line 127-132: delete authors names

Reply: Thank you for this suggestion; however, we decided to keep this text format because it is largely used in Foods journal.

Line 145-143: It could be interesting to include the effect of these by-products on protein oxidation and metmyoglobin formation.

Reply: A sentence has been added, as suggested (please see page 5, lines 160-165).

Line 145: It could be interesting to include a brief section where the parameters to evaluate meat quality are included or what they consist of, so that the reader who is not an expert in the subject, can follow up on the results mentioned later in the manuscript

Reply: Thank you for this comment. We attempted to clarify this point in the revised version of the manuscript (please see page 5, lines 167-171).

Line 193: the table must appear immediately after the first paragraph in which it is mentioned

Reply: The reviewer is absolutely right. Table 2 is now placed at the end of the first paragraph in which it is mentioned (please see page 5, line 177).

Line 152: indicate the meaning of a* and b* parameters

Reply: Thank you for this suggestion. This information is now available in page 6, line 183.

Line 156,163,166: delete authors names

Reply: Again, the authors do not agree with the reviewer and they would like to maintain this text format.

Line 157: modify…on SFA, MUFA and PUFA.

Reply: Corrected, as suggested by the reviewer.

Line 157: modify…28 days by 28-d or 28 d; abbreviate from this section throughout the document

Reply: Corrected, as suggested by the reviewer.

Line 164:…but increased PUFA

Reply: Corrected, as suggested by the reviewer.

Line 174,175,178,182: delete authors names

Reply: As mentioned before, we would like to maintain the text format.

Line 176: indicate the meaning of MDA

Reply: We apologize for this. The meaning of MDA is now added at the first time that it appears in the revised manuscript (please see page 6, line 185). In addition, the authors removed the sentence “Gray and Pearson suggested 1 mg of MDA/kg in the muscle as a threshold for the organoleptic detection of rancidity” since it is out of the context.

Line 184: indicate the meaning of L* parameter

Reply: Corrected, as suggested by the reviewer.

Line 193: it is necessary delete the authors names of references included in the table

Reply: The authors agree with the reviewer. The authors names in the tables are now deleted, as requested.

Line 193: in the reference Yan and Kim [62],…in which percentage the  color values were increased, as well as the reduction in the oxidation of lipids respect to the control

Reply: The percentages of increase or decrease of color parameters and lipid oxidation were added in the revised manuscript. The reference Yan and Kim is now [68] instead of [62].

Line 193: in the references Habeanu et al. [64] and Bertol et al. [68], the animal age was not indicated

Reply: The age of the animals was not reported in both studies. The reference Habeanu et al. is now [69] instead of [64] and Bertol et al. [70] instead of [68].

Line 193: in the reference Bertol et al. [68], modify…..(3–5%)..(6–10%)

Reply: Corrected, as suggested by the reviewer.

Line 193: in the reference Kafantaris et al. [65], the animal weight was not indicated

Reply: The animal weight is now indicated in the revised document and the reference number changed to [71].

Line 193: in the references [64,65,67], by what percentage did the values increase or decrease?

Reply: The percentages of increase or decrease in the cited references are now added in the revised manuscript. The reference numbers [64,65,67] were changed to [69,71,72].

 Line 193: in the references [67], indicate the meaning of WHC, and indicate the sensory traits

Reply: Thank you for this correction. The meaning of water holding capacity (WHC) appears now in page 5, line 168 before Table 2. The sensory attributes evaluated (tenderness, juiciness and off-flavours) are now described in Table 2.

Line 193: in the reference [50], the animal age was missed and indicate the percent increase respect to the control

Reply: The age of pigs was not indicated in the cited study. In addition, the incorporation of 5% flax meal and 1% grape seeds meal in pigs’ diet promotes an increase of 38% in n-3 PUFA (2.12% of total fatty acids) relative to the reference diet (1.32% of total fatty acids), because flax meal is rich in n-3 PUFA. The reference number [50] is now [31].

Line 193: in the reference [70], the animal age was missed

Reply: The age of pigs was not indicated in the cited study. The reference number is now [73] in the revised manuscript.

 Line 193: in the reference [71], the animal age was missed and indicate the percent increase or decrease respect to the control

Reply: The age of finishing pigs was not indicated in the study by Xu et al. In addition, the supplementation with grape seed proanthocyanidin extract (GSPE) in pigs diet significantly (p < 0.05) increased pH24h, a* value and crude protein content (3%, 15% and 7%, respectively) compared to the control. Moreover, the dietary supplementation with GSPE resulted in greater PUFA, n-3 PUFA contents and PUFA/SFA ratio (20%, 13% and 26%, respectively).  In contrast, compared to control, drip loss at 48h and shear force decreased by 39% and 4%, respectively, with dietary supplementation of GSPE. The reference number [71] is now [74].

Line 199: delete authors names; it is necessary to make changes through the manuscript

Reply: As mentioned before, we would like to maintain the text format.

 Line 200:… muscle colour (L* and b*), but….

Reply: Corrected, as suggested by the reviewer.

Line 201: …decrease a* values.

Reply: Corrected, as suggested by the reviewer.

Table

Line 212: [73–75,77–79].

Reply: Corrected, as suggested by the reviewer. The references number are now [78–82,86] in the revised manuscript.

Line 225: …5 wk also…

Reply: Corrected, as suggested by the reviewer.

Line 234: indicate the percentage of increase or reduction in the values that presented a significant effect with respect to the control

Reply: Thank you for this suggestion. The percentages of increase or reduction are now added in the revised manuscript. However, the percentage of redness decrease is only an approximate value (reference number [29]), because breast colour changes in the cited work are presented in Figure.

Line 234: in some references the weight of the animals is omitted, could you specify if the value was not reported in the cited reference, if it is not available

Reply: In some references in Table 3, the weight of the animals is omitted because the value was not reported in the cited works.  

Line 234: in the reference [74], pH48h

Reply: Corrected, as suggested by the reviewer. The reference number changed to [79].

Line 248: 15638–15678.

Reply: Corrected, as suggested by the reviewer.

Line 287: 217–260.

Reply: Corrected, as suggested by the reviewer.

Line 293: 221–237.

Reply: Corrected, as suggested by the reviewer.

Line 297: 152–157.

Reply: Corrected, as suggested by the reviewer.

Line 302: 1–17.

Reply: Corrected, as suggested by the reviewer.

Line 304: 11579–11587.

Reply: Corrected, as suggested by the reviewer.

Line 306: 1607–1615.

Reply: Corrected, as suggested by the reviewer.

Line 308: 267–337.

Reply: Corrected, as suggested by the reviewer.

Line 310: 2473–2505. ….review the page range format in the following references

Reply: Corrected, as suggested by the reviewer.

Reviewer 3 Report

The authors have submitted an interesting systematic review article on the use of grape by-products to enhance meat quality and nutritional value in monogastrics, which are of great importance to primary producers, meat industry and consumers.

My few comments and suggestions are listed below.

Line 176: MDA/kg - Please, write in full on the first occurrence.

Line 178:  O’Grady and colleagues [70] - O’Grady et al. [70]

Table 2 and 3. Column with references - Please, put only numbers of references in this column.

Line 199: Kasapidou and colleagues [48] - Kasapidou et al. [48]

Line 212: [73,74,75,77,78,79]. - [73-79]

Author Response

Comments and Suggestions for Authors

The authors have submitted an interesting systematic review article on the use of grape by-products to enhance meat quality and nutritional value in monogastrics, which are of great importance to primary producers, meat industry and consumers.

My few comments and suggestions are listed below.

Line 176: MDA/kg - Please, write in full on the first occurrence.

Reply: Corrected, as suggested by the reviewer.

Line 178:  O’Grady and colleagues [70] - O’Grady et al. [70]

Reply: Corrected, as suggested by the reviewer. The reference number is now [73].

Table 2 and 3. Column with references - Please, put only numbers of references in this column.

Reply: Corrected, as suggested by the reviewer.

Line 199: Kasapidou and colleagues [48] - Kasapidou et al. [48]

Reply: Corrected, as suggested by the reviewer. The reference number is now [29].

Line 212: [73,74,75,77,78,79]. - [73-79]

Reply: Corrected, as suggested by the reviewer. The references number are now [78–82,86] in the revised manuscript.

Round 2

Reviewer 2 Report

I do not have any comments, all observations were taken care of